# A Comprehensive Study of MicroRNA in Baculoviruses

**DOI:** 10.3390/ijms25010603

**Published:** 2024-01-03

**Authors:** Lucas Federico Motta, Carolina Susana Cerrudo, Mariano Nicolás Belaich

**Affiliations:** Laboratorio de Ingeniería Genética y Biología Celular y Molecular—Área Virosis de Insectos, Instituto de Microbiología Básica y Aplicada, Departamento de Ciencia y Tecnología, Universidad Nacional de Quilmes, Roque Sáenz Peña 352, Bernal B1876BXD, Buenos Aires, Argentina; lmotta.ligbcm@gmail.com

**Keywords:** baculovirus, microRNA, RNA folding, gene regulation

## Abstract

Baculoviruses are viral pathogens that infect different species of Lepidoptera, Diptera, and Hymenoptera, with a global distribution. Due to their biological characteristics and the biotechnological applications derived from these entities, the *Baculoviridae* family is an important subject of study and manipulation in the natural sciences. With the advent of RNA interference mechanisms, the presence of baculoviral genes that do not code for proteins but instead generate transcripts similar to microRNAs (miRNAs) has been described. These miRNAs are functionally associated with the regulation of gene expression, both in viral and host sequences. This article provides a comprehensive review of miRNA biogenesis, function, and characterization in general, with a specific focus on those identified in baculoviruses. Furthermore, it delves into the specific roles of baculoviral miRNAs in regulating viral and host genes and presents structural and thermodynamic stability studies that are useful for detecting shared characteristics with predictive utility. This review aims to expand our understanding of the baculoviral miRNAome, contributing to improvements in the production of baculovirus-based biopesticides, management of resistance phenomena in pests, enhancement of recombinant protein production systems, and development of diverse and improved BacMam vectors to meet biomedical demands.

## 1. Baculovirus Biology

*Baculoviridae* is one of the most extensively studied viral families, despite not infecting humans or animals and plants of significant socio-economic interest. Instead, it primarily infects various species of arthropod larvae, including Lepidoptera, Hymenoptera, and Diptera, many of which are considered agricultural pests [1]. This explains the considerable interest in baculoviruses, as they have been successfully utilized as the main active ingredients in bioinsecticide formulations for controlling insect pests in numerous agricultural crops [2]. Over time, as our understanding of baculoviruses has grown, their primary biotechnological application has diversified into other industries, resulting in the development of genetically modified variants that serve as baculoviral recombinant protein expression vectors (*Baculovirus Expression Vector Systems*, or BEVS) [3]. Subsequently, a new type of application emerged involving engineered virions with utility in mammals. Although these virions are non-infectious in vertebrates, they can transduce mammalian cells and carry recombinant DNA of interest, facilitating the conceptual development of vaccines and therapies (referred to as BacMam applications for *Baculovirus applied in mammals*). These attributes position it as a technological competitor to adenoviral vectors, offering similar benefits but with a superior immunological profile [4,5].

Baculoviral nucleic acid is a large circular dsDNA molecule that resides within a proteinaceous polar nucleocapsid. The family derives its name from the characteristic morphology of this nucleocapsid, which is surrounded by a lipid bilayer. These structures are known as occlusion-derived virions (ODVs) and are embedded in a proteinaceous paracrystalline matrix called occlusion body (OB), which forms in the nucleus of the infected insect cell [1]. Among the described baculoviruses, two main types of OBs have been reported: granuloviruses (GVs), with a granular morphology, and nucleopolyhedroviruses (NPVs), with polyhedral forms. GVs typically contain a single ODV, while NPVs can contain multiple ODVs. Some NPV isolates even have ODVs with more than one nucleocapsid per lipid envelope [6]. This morphological diversity, coupled with host specificity and other attributes, has led to their classification into four genera: *Alphabaculovirus* (NPVs infecting Lepidoptera); *Betabaculovirus* (GVs infecting Lepidoptera); *Deltabaculovirus* (NPVs infecting Diptera); and *Gammabaculovirus* (NPVs infecting Hymenoptera) [7,8]. Currently, *Baculoviridae* is a family included in the order *Lefavirales*, which also includes *Nudiviridae* and *Hytrosaviridae*, within the Class *Naldaviricetes* (which incorporates *Nimaviridae*) [9,10]. It is worth noting that there is another viral morphotype called budded virion (BV), which is generated during the infection process, at least for alphabaculoviruses and betabaculoviruses [1]. BVs facilitate the spread of the infection to different tissues of the affected larva. BVs are nucleocapsids enveloped by a lipid membrane containing viral proteins, distinct from the membrane surrounding ODVs. With these two types of virions, the infective cycle of baculoviruses is typically divided into a primary process governed by OBs and ODVs in the midgut cells of susceptible larvae and a secondary process directed by BVs (if present), which can disseminate the infection to most tissues of the affected individual [11]. The pathology induced by baculoviruses is typically lethal and may involve larval liquefaction as a possible outcome [1].

The intricate infective cycle of baculoviruses is made possible by the sophisticated regulation of their genetic content. The International Committee on Taxonomy of Viruses (ICTV; https://ictv.global/taxonomy, last accessed on 20 September 2023) proposes the existence of 103 species of baculovirus based on genetic distance among core genes and other biological properties such as virion morphology and host impacts. A recent study, which considered all available baculoviral genome data, proposed a similar outcome solely based on genetic distance criteria. This study explored sequences ranging between 81,755 and 178,733 base pairs and encompassed between 89 and 183 protein-coding genes [12]. Baculoviruses possess their own machinery for transcription and replication, and all reported members share at least 39 orthologous protein genes [12,13,14]. Interestingly, despite the extensive protein gene content in baculoviral genomes, the presence of non-coding RNA genes has also been observed, including those that encode microRNAs (miRNAs) [15] and even transfer RNAs (tRNAs) [16]. Based on the reported studies, there is evidence of informational overlap within baculoviral genomes, where both protein-coding and non-coding RNA molecules may be encoded in the same genomic locus [17]. Additionally, non-genic functions, such as replication origins, may also be present [18,19]. A more comprehensive understanding of the function of each protein gene, as well as the syntax and role of RNA genes, will not only contribute to the understanding of interactions between hosts and their parasites in nature but also hold potential implications for human applications derived from the use of these viruses, including their natural and engineered variants.

## 2. miRNA Biogenesis and Function

Numerous cellular processes in both prokaryotes and eukaryotes are regulated by enzymatic complexes composed of RNA and protein molecules. These complexes, known as ribonucleoproteins (RNPs), typically act on nucleic acid substrates. They play various critical roles, such as facilitating transcript maturation through alternative splicing [20], regulating gene expression by selectively degrading or inhibiting the translation of transcripts [21], and serving as specific nucleases to eliminate foreign nucleic acids [22]. Additionally, there are RNPs involved in nuclear chromatin remodeling, which have significant epigenetic consequences [23]. RNA molecules within these RNPs have pivotal roles as they often serve as guides, directing the enzymatic machinery to their target nucleic acids.

The biogenesis of these RNA molecules occurs through the transcription of genes that typically lack protein-coding information. These RNA genes can generate, for example, small nuclear RNAs (snRNA), microRNAs (miRNAs), or CRISPR (Clustered Regularly Interspaced Short Palindromic Repeats) RNAs (crRNAs) that participate in RNP complexes known as spliceosomes, RNA-induced silencing complexes (RISCs), or CRISPR/Cas systems. Long non-coding RNAs (lncRNAs), which are abundant in eukaryotic regulatory dynamics, also form complexes that specifically bind enzymes at certain loci in the genome [24]. For any of these examples, the annotation of RNA genes in genomes is usually more challenging compared to protein genes because our knowledge of the syntactic elements that comprise them is somewhat limited. The absence of an open reading frame (ORF), although possibly interrupted by introns, is no longer a hindrance to its complete identification. As a result, some of the processes occurring in cells may be overlooked or not fully explained, with these RNA molecules serving as a type of “dark matter” that could contribute to a better understanding of many mechanisms that shape and support organisms.

RNA interference (RNAi) mechanisms are cellular processes that have been recognized for about 25 years and have been successfully described biochemically [25], enabling the design and development of biotechnologies for agriculture [26] and medicine [27]. This highly conserved process among eukaryotes [28] relies on RNPs that target RNA molecules. The mechanism is initiated by the presence of double-stranded RNA (dsRNA produced endogenously or derived from invading nucleic acids), which triggers the formation of RNPs with nuclease activity or translation inhibition, allowing the cell to defend itself against viral attacks and enriching its regulatory mechanisms for gene expression. Considering this last activity, the RNAi mechanism may intervene as post-transcriptional gene silencing (PTGS), providing cells with options to adapt their functions according to the dynamic signals they receive. This conserved function is a complex system encoded from protein-coding genes involved in dsRNA processing and the formation of RNPs, which will fulfill the role of effector nucleases or translation blockers [29]. Additionally, it includes RNA genes, such as the miR genes that produce miRNAs, which generate the guides that lead the RNPs to the target molecules [30]. Among the protein-coding genes, it is worth mentioning those that encode the type III RNases Drosha [31] and Dicer [32], DGCR8 (DiGeorge syndrome critical region 8, in humans), or Pasha (partner of Drosha in non-humans) [33] as well as the H-type RNases known as Argonauts (AGO) [34]. Since the beginning of the century, thousands of miRNAs from various organisms have been described and stored in repositories [35], and there continues to be scientific interest in the role of these biomolecules in the physiology and pathology of living beings (Figure 1).

While most scientific efforts are focused on understanding mammalian miRNAs, especially those involved in human processes, there are also reports on insects and viruses. These contributions enrich our global understanding of the RNA interference process and can potentially create new opportunities for advancing various biotechnologies.

Generally, the proteins involved in this silencing function are constitutively expressed without significant modulation, while the regulatory molecules (such as miRNA) provide the inducible action. Therefore, it is crucial to characterize miR genes and understand their expression dynamics to comprehend the occurrence of silencing processes. MiR genes are typically delimited by RNA polymerase II promoter and termination signals (although there may be other origins, such as its occurrence in introns [36]), giving rise to a primary precursor (pri-miRNA) of approximately 70–100 nucleotides [37]. This molecule usually adopts a hairpin-like structure, recognized in the nucleus by Drosha/Pasha (microprocessor complex), a protein that processes it to generate pre-miRNA [33]. Subsequently, the processed transcript is exported to the cytoplasm by Exportin 5/RanGTP [38], where it is cleaved by Dicer with the support of other RNA-binding proteins such as TRBP (the transactivating response RNA-binding protein) in humans, leading to the formation of RISC containing mature miRNA and AGO proteins [21]. This mature miRNA, typically 21–25 nucleotides long, is derived from the double-stranded region of the pri-miRNA (Figure 2).

When the RISC complex encounters target transcripts (usually mRNA) through specific interactions, it proceeds to “silence” the target. This can involve hydrolysis or inhibition of translation, resulting in an “OFF” state for the gene product being targeted. To carry out this action, an RNA–RNA interaction (miRNA–mRNA) needs to occur, typically involving specific sequences located in the 3′ untranslated region (UTR) of the target molecule. This region, known as the miRNA response element (MRE), acts as a regulatory signal for gene expression [39]. It has been observed that the conformation of the RISC complex is determined by the specific binding between the RNP and the target RNA, involving A:U and G:C base pair interactions, as well as the G:U pair. Specifically, this interaction is facilitated by a sequence located approximately 6 to 8 nucleotides from the 5′ end of the miRNA, referred to as the core or seed sequence [40]. This allows for the prediction of potential target genes in a genome that could be regulated by a given miRNA.

## 3. Baculovirus-Encoded miRNAs

Viruses are parasitic entities that utilize cellular resources from the host cells they invade in order to replicate and sustain themselves in nature. Within this complex relationship between host and parasite, molecular dialogues occur that aim to either terminate the infective process or weaken the host’s defense mechanisms to facilitate successful invasion. RNAi mechanisms play a significant role in these interplays within eukaryotes [41,42]. As a result, the presence of cellular miRNA genes that can impact viral processes, as well as viral miRNA genes that regulate their own expression cascades to counteract the host, emerge as notable features in viral infections. 

Under this context, baculoviruses are no exception. As mentioned earlier, their genomes contain miR (bac-miR) genes. A comprehensive literature review identifies 13 reported bac-miRs with experimental evidence (Table 1), focusing on them from now on. Among them, five have been found in Bombyx mori nucleopolyhedrovirus (BmNPV; *Aplphabaculovirus bomori*), five in Autographa californica multiple nucleopolyhedrovirus (AcMNPV; *Alphabaculovirus aucalifornicae*), one in Anticarsia gemmatalis multiple nucleopolyhedrovirus (AgMNPV; *Alphabaculovirus angemmatalis*), and two in Spodoptera litura nucleopolyedrovirus (SpltNPV; *Alphabaculovirus spliturae*). It is worth mentioning that all these viruses belong to the *Alphabaculovirus* genus. This does not necessarily imply that the occurrence of bac-miR genes is exclusive to this clade, but rather, more research efforts have been directed here due to factors such as the greater availability of tools, like existing insect cell lines susceptible to these viruses.

The reported bac-miRs overlap with protein-coding regions, some of which have orthologs (core genes) in all known baculoviral genomes [12], including bmnpv-miR-3, AcMNPV-miR-1, AcMNPV-miR-3, AcMNPV-miR-5, agmnpv-miR-4, and Splt-NPV-miR-11698-3p. The remaining bac-miRs (except for bmnpv-miR 4) are not only located within alphabaculovirus coding genes but also within those of other genera. The fact that they are situated within sequences conserved throughout the evolution of *Baculoviridae* may suggest that bac-miRs are underannotated as baculoviral genes. Given this, it would be intriguing to validate this assumption by conducting both bioinformatics and experimental investigations on the presence of bac-miRs homologous to those reported in orthologous protein genes of other baculoviral species.

Moreover, their target RNA molecules would include not only viral mRNAs but also those of the host, thereby facilitating control over both their own gene expression and that of the parasitized organism. In some cases, experimental evidence of their targets has not been demonstrated. In a similar manner to the loci where bac-miRs are identified, the reported target RNAs consist of transcripts derived from protein-coding genes with high conservation in *Baculoviridae*, including many core genes such as *DNA polymerase*, *p6.9*, *p40*, *p95*, *vlf-1*, *lef-8*, *DNA helicase*, *odv-e25*, *ac95*, *ac101*, *ac66,* and *ac98*. This further suggests the significance these RNA genes might hold across the entire family rather than being limited to the species in which they have been described.

Furthermore, the transcriptional machinery responsible for their production has not been fully elucidated, which hinders the prediction of typical promoter motifs and downstream polyadenylation signals for the identified pre-miRNAs (Table 2). For certain bac-miRs and assuming the limits composed by the hypothesized motifs in this work, the pri-miRNA sequences could be relatively long, which aligns with observations made in other biological systems.

Based on our literature search, the 13 bac-miRs collectively target a total of 18 host mRNA molecules (although only three of them have experimental validation). To determine if these genes are associated with specific underlying molecular pathways and functional categories, such as Gene Ontology (GO), a gene list enrichment analysis was conducted in this study using the ShinyGO server (http://bioinformatics.sdstate.edu/go/; last accessed on 27 October 2023; [43]). This allows us to identify 31 functionally enriched categories with a significance threshold of FDR (false discovery rates) <0.05 (descriptive details are provided in Appendix A). Eight of these categories pertain to biological processes, while the remainder relate to molecular functions. These categories primarily involve metabolic and cellular processes, with three also participating in biological regulation (positively regulating gene translation and expression, as well as cellular amide metabolic processes). Specifically, these processes include lipid and organic acid metabolism, nitrogen compound metabolism, cellular respiration, electron transport chain metabolic processes, and organic substance biosynthetic processes. Notably, all significantly enriched molecular functions are associated with catalytic activities or binding to compounds that are integral to the aforementioned biological processes.
ijms-25-00603-t001_Table 1Table 1Baculovirus miRNA with experimental evidence. The order of the bac-miRs in the first column follows the historical sequence of scientific publications proposing them for each viral species involved. It separates those belonging to Group I Alphabaculovirus (BmNPV, AcMNPV, AgMNPV) from those belonging to Alphabaculovirus Group II (SpltNPV). In the fourth and fifth columns, it is indicated in parentheses in which genera orthologs for that gene are found (A = *Alphabaculovirus*; B = *Betabaculovirus*; G = *Gammabaculovirus*; D = *Deltabaculovirus*). In cases where this information is presented in bold, it indicates that orthologs are identified in all species within the specified genera (core gene for that group of genera). In the fourth column, the “c” in superscript indicates that bac-miR is located on the complementary strand to the protein-coding gene. In the fifth and sixth columns, the underlined genes reference that there is experimental evidence for such an assignment. This table is original and has not been previously published.Bac-miRVirusHostLocusViral TargetHost TargetReferencebmnpv-miR-1BmNPVBombycidaeCathepsin (AB) ^c^bro-I (ABD), fusolin (ABG), DNA polymerase (ABGD)GTP-binding nuclear protein Ran, serine protease[44,45]bmnpv-miR-2BmNPVBombycidaeChitinase (AB) ^c^Chitinase (AB)Eukaryotic translation initiation factor 4E-binding protein, Bombyx mori Hemolin[44]bmnpv-miR-3BmNPVBombycidaelef5 (ABGD) ^c^bro-III (AB), p6.9 (ABGD), p40 (ABGD), p95 (ABGD), vlf-1 (ABGD), Fusolin (ABG), chitinase (AB)Prophenoloxidase[44,46]bmnpv-miR-4BmNPVBombycidaevp80 (A)lef-8 (ABGD), p25 (A+B)_C_, helicase (ABGD), ORF3(AB)Chymotrypsin-like serine protease, cadherin-like membrane protein, DEAD box polypeptide[44]BmNPV-miR-415BmNPVBombycidaeodv-e66 (AB)-Rapamycin isoform 2 (ToR2)[47]AcMNPV-miR-1AcMNPVNoctuidaeodv-e25 (ABGD) ^c^ac18, odv-e25 (ABGD), ac95 (ABGD), ac15 (AB), ac131 (ABG), ac82 (AB), ac34(AB), ac101 (ABGD), ac55(AB), ac135 (AB), ac30, ac66 (ABGD), ac11(A), ac64 (AB)-[48,49]AcMNPV-miR-2AcMNPVNoctuidaelef-6 (AB) ^c^lef-6 (AB), lef-11 (ABG), orf-49 (A), orf-63, orf-38 (AB)-[17]AcMNPV-miR-3AcMNPVNoctuidaeodv-C42 (ABGD) ^c^ac10 (AB), f-protein (ABD), ac25 (ABG), ac86, ac98 (ABGD), ac101 (ABGD), chitinase (AB)-[50]AcMNPV-miR-4AcMNPVNoctuidaecg30 (AB)-Alg-2, rop[17]AcMNPV-miR-5AcMNPVNoctuidae49k (ABGD)Twenty predicted candidate targets without evident regulation; further experimental investigation is required[17]agmnpv-miR-4AgMNPVNoctuidaep48 (ABGD) ^c^lef-8 (ABGD)-[51]SpltNPV-miR-11684-3pSpltNPVNoctuidaepk1 (AB)–hoar (AB) ^c^-V-type proton ATPase catalytic subunit A-like (vATPaseA), replication factor C4 (repC4), NADH dehydrogenase[ubiquinone] 1 subunit C2-like (ndc2)[52]SpltNPV-miR-11698-3pSpltNPVNoctuidaep6.9 (ABGD) ^c^-Eukaryotic initiation factor 4A-like isoform 1 (eIF4A), prostaglandin reductase 1 (pgr1), H2A histone family member V (H2A), FAD-dependent oxidoreductase[52]
ijms-25-00603-t002_Table 2Table 2Baculovirus miRNA sequence details. The genome GeneBank accession used are BmNPV (L33180.1), AcMNPV (NC_001623.1), AgMNPV (NC_008520.2), SpltNPV (NC_003102.1). In the third and fourth columns, the letter “c” indicates that the sequence is in the complementary strand. In the fourth column, the identity of the ORF is found in Table 1. The potential promoter and terminator signals, as indicated in the fifth and sixth columns, are referenced to the boundaries of the pre-miRNAs and are predictive bioinformatic determinations made by this work. This table is original and has not been previously published.Bac-miRSequence miRNALocus Pre-miRNA (Mature miRNA)ORF LocalizationPutative Promoter  (Genomic Position)Putative PolyA Signal  (Genomic Position)bmnpv-miR-1AAAUGGGCGGCGUACAGCUGGc99262–99347 (c99318–99338)98756–99727CAGT(−79), TAAG(−537)ATTAAA(+426)bmnpv-miR-2GGGGUUUUUGUACGGCGGCCC98016–98105  (98020–98040)c97049–98707TATA(−374), TAAG(−454)ATTAAA(+459)bmnpv-miR-3GAAAGCCAAACGAGGGCAGGCGc80268–80356 (c80278–80299)79558–80355CAGT(−64)AATAAA(+1969)bmnpv-miR-4GGUGGAUGUGAUUGUUGACGACA83292–83385  (83308–83330)83240–85318TATA(−59), TAAG(−89)ATTAAA(+58)BmNPV-miR-415UCGAGUUGACGGCCGUGGGCc33600–33463 (c33522–33541)32866–34974TATA(−122), TAAG(−149)ATTAAA(+554)AcMNPV-miR-1GCGACGACUCGGUUAAGGAAc80100–80042 (c80045–80064)79971–80657TATA(−93), CAGT(−318)AATAAA(+20)AcMNPV-miR-2AACACCUGCCUGUAGGCGCGCUUc23672–23599 (c23603–23625)23465–23986CAGT(−169), TATA(−348)ATTAAA(+70)AcMNPV-miR-3GCGGCGUAGGCUGCGCGGACGCU87727–87793  (87733–87755)c86921–88006CAGT(−213)AATAAA(+856)AcMNPV-miR-4UUUCUGCAACCAAUAGACAGAc75504–75450 (c75451–75471)c74737–75531CAGT(−88), TATA(−129), TAAG(−316) ATTAAA(+162)AcMNPV-miR-5UAGACGAUGCCGUGCUCAUGAA123775–123866 (123783–123804)123632–125065TAAG(−161), TATA(−167)AATAAA(+182)agmnpv-miR-4GGUGGCCGCCACUUUGUUUACUUc49475–49370 (c49450–49472)48600–49826TATA(−267)AATAAA(+34)SpltNPV-miR-11684-3pGGUCGUGUCUUCCACUUCCUU3434–3522  (3486–3506)2392–3204  c3519–5714TATA(−39), CAGT(−143), TAAG(−220)AATAAA(+360)SpltNPV-miR-11698-3pUCGAGCGGCGAUGGUAGAAG87923–88008  (87974–87993)c87850–88104TATA(−190)AATAAA(+135)

## 4. Shared Characteristics among Described bac-miRs

In a similar manner to open reading frames in protein-coding genes, which can uncover compositional biases related to codon distribution aiding in predicting their existence, miR genes can also exhibit compositional biases that aid in their prediction. Therefore, the identified bac-miRs can serve as a training dataset to uncover shared characteristics such as loop occurrence, stem length, and stability. These parameters can provide valuable insights for refining prediction algorithms. Regarding this, the described bac-miR was first characterized in this study in terms of length and richness in GC (Figure 3).

This analysis revealed a significant dispersion in GC content, both in the pre-miRNA and its mature forms, although values close to and even higher than 50% are predominant, especially in mature variants. It is worth noting that when calculating the ratio between the percentage of GC content in the mature miRNAs and the genomes that harbor them (R = % GC miRNA/% GC genome), a value of 1.51 +/− 0.33 is observed, indicating a distinctive bias in this attribute. On the other hand, the length is shown to be more conserved, with a length between 55 and 138 for pre-miRNAs (90.5 +/− 16.05) and between 20 and 23 for miRNAs (21.6 +/− 1.03). These values are consistent with those reported in other biological systems, and they may be useful for the definition of predictive algorithms for bac-miRs. 

Another question to be addressed is whether there are common patterns in the secondary structures of pri-miRNAs, which serve as substrate molecules for Drosha/Pasha and then for DICER (Figure 4). The analysis of these characteristics could reveal whether there are conserved lengths in the double-stranded regions and their stability.

The results obtained in this work (which are very similar to those previously published) show potential shared structural biases. To validate this, our investigation was further enhanced by assessing the extent of structural conservation in baculoviral pre-miRNAs (Figure 5). This enabled us to identify that 11 out of the 13 baculoviral pre-miRNAs could be categorized into two groups based on their shared structures despite originating from distinct viral species. Notably, the pre-miRNAs in Group 1 possess a terminal loop and a long stem interrupted by a large inner loop, whereas those in Group II display a terminal loop and a stem with four small inner loops.

Another interesting analysis to conduct involves predicting the end of the double-stranded RNA derived from the pri-miRNA (pre-miRNA molecule) with lower stability at its 5′ end. This attribute is considered a crucial factor in determining which strand will serve as the passenger and which as the guide (mature miRNA) in the conformation of RISC [56]. As such, the theoretical values of thermodynamic stability at these endpoints were calculated (Table 3).

The obtained results revealed that six bac-miRNAs (bmnpv-miR-1, bmnpv-miR-2, AcMNPV-miR-1, AcMNPV-miR-2, AcMNPV-miR-4, and SpltNPV-miR-11684-3p) exhibit a less stable ΔG at the 5′ end corresponding to the mature miRNA, either in the external loop or in the initial stack. Meanwhile, three miRNAs showed a less stable or equally stable ΔG only in the first stack (BmNPV-miR-415, AcMNPV-miR-3, and SpltNPV-miR-11698-3p). Only four molecules display 5′-end stability features that contradict the expected trend. Despite the challenges in conducting these predictions of thermodynamic stability of the ends (due to uncertainty regarding the identity of the small double-stranded RNA composed of the guide and passenger strand), and although it was not observed that in all cases, the less stable 5′ end gives rise to the mature miRNA, a positive trend toward the conservation of this attribute is detected.

## 5. Bioinformatics Tools for miRNA Studies

The analysis shown in this work on the bac-miRs reported in alphabaculoviruses plus other previously published integrative studies [58] provides useful insights to guide future predictions of novel non-coding genes that express miRNA molecules in baculovirus genomes. To further enhance the study in this area, it is noteworthy to emphasize the existence of numerous databases and software tools that assist in the miRNA characterization and target search. Accordingly, some of them are presented (Table 4).

Effective integration of the previously established attributes in the reported bac-miRs, coupled with the judicious use of the demonstrated tools and the customized design of computational routines to interconnect them into a robust pipeline, holds the potential to predict previously undocumented bac-miRs. This represents a pivotal initial step for subsequent experimental validation of these putative non-coding genes. It contributes significantly to our comprehension of the regulatory dynamics of gene expression in baculoviruses and further enriches our understanding of the molecular interplay that occurs between the parasite and its host during the infection process.

## 6. Concluding Remarks

Considering the compiled literature and the conducted analyses, it can be concluded that the 13 bac-miR genes are located within protein-coding regions in evolutionarily conserved sequences within *Baculoviridae*, many of them with orthologs in more than one genus. They give rise to primary transcripts that generate pre-miRNA of approximately 90 nucleotides (in a range between approximately 50 and 150 nucleotides), which exhibit at least two distinct conformational patterns (referred to in this work as Groups 1 and 2), leading to the production of miRNA molecules around 20–23 nucleotides in length. These miRNA molecules are characterized by a higher GC content than their unprocessed primary transcripts and that of the genome in which they are found. The targeted mRNAs comprise both viral genes, most of which have orthologs present across all species within the same genus and cellular genes involved in cellular and metabolic processes. Integrating these analyses made on the reported bac-miRs with experimental demonstrations could serve as a foundation for predicting undetected bac-miRs. This step could precede their confirmation through transcriptomic and functional evidence.

## Figures and Tables

**Figure 1 ijms-25-00603-f001:**
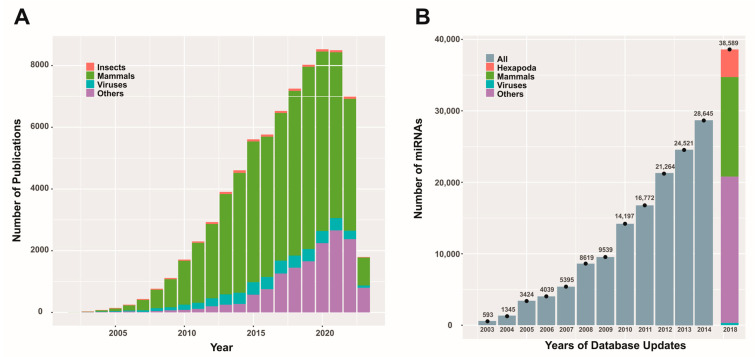
Scientific publications and miRBase database content related to miRNAs. Panel (**A**) depicts the number of publications (original articles) per year on miRNAs from different entities (except for 2023, where the results are partial). The data were obtained using the PubMed search (https://pubmed.ncbi.nlm.nih.gov/; last accessed on 3 April 2023) with the keyword “microRNA” and corresponding combinations for the entities considered. Panel (**B**) presents a bar graph illustrating the reported miRNA quantities in different updates of the miRBase database (https://www.mirbase.org/; last accessed on 3 April 2023) for various entities (when this information was available). This figure is original and has not been previously published.

**Figure 2 ijms-25-00603-f002:**
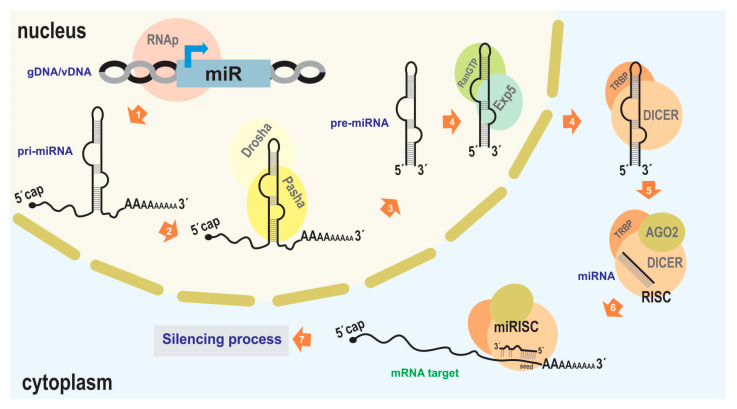
miRNA-mediated silencing mechanism. The illustration depicts the canonical pathway of endogenous miRNA-mediated silencing. 1. Transcription from the cellular genome (gDNA) or a viral nucleic acid (vDNA) to generate the pri-miRNA (usually carried out by RNA polymerase II). 2, 3. Association with Drosha and Pasha proteins (DGCR8 in humans) to generate the pre-miRNA. 4. Export to the cytoplasm facilitated by Exportin5 and RanGTP. 5. Association with DICER and TRBP, which, along with the presence of Argonaut 2 (AGO2), complete miRNA maturation and form the RNA-induced silencing complex (RISC). 6. Binding of the miRISC to the target transcript (typically in the 3′ UTR of an mRNA). 7. Degradation of the target transcript or inhibition of its translation; reduction in target RNA molecules and/or a decrease in the production of the encoded protein (in the case of mRNA), leading to subsequent effects on biological functions. As the protein machinery involved in RNA interference in the “baculovirus/insect” system is derived from host genes, GenBank accessions of the orthologs of the main proteins are indicated in some of the lepidopterans that host baculoviruses where bac-miRs were reported: NP_001095931.1 (Ago1), NP_001036995.2 (Ago2), NP_001180543.1 (DICER-2), XP_012547603.2 (Drosha), XP_037872173.1 (Pasha), of *Bombyx mori*; XM_022976077.1 (AGO), AHC98010.1 (AGO2), XM_022972435.1 (DICER), XP_022826597.1 (Drosha), XP_022831748.1 (Pasha), of *Spodoptera Litura*; AVK59452.1 (AGO1), XP_035453543.1 (AGO2), XP_035450082.2 (DICER), XP_050556783.1 (Drosha), XP_050553060.1 (Pasha), of *Spodoptera frugiperda*. This figure is original and has not been previously published.

**Figure 3 ijms-25-00603-f003:**
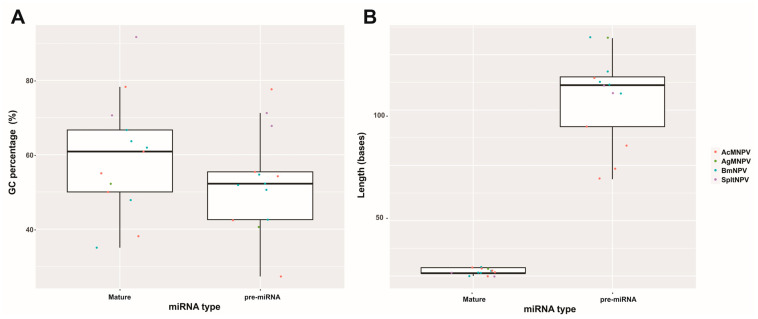
Bac-miR common sequential features. Box plot showing the GC percentage (**A**) and length (**B**) of the baculoviral pre-miRNAs and mature miRNAs, generated using an ad hoc script implemented with Biopython [53]. For these analyses, the sequences reported in publications where the 13 bac-miRs are described were considered. This figure is original and has not been previously published.

**Figure 4 ijms-25-00603-f004:**
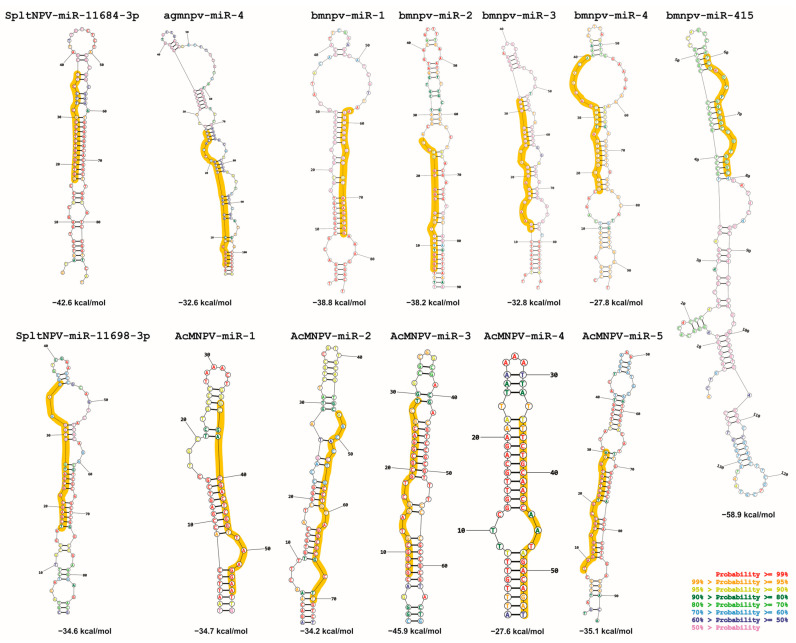
Secondary structures of baculoviral pre-miRNAs. The secondary structures of reported experimental pre-miRNAs from AcMNPV, BmNPV, AgMNPV, and SpltNPV are depicted along with their corresponding minimum free energy (MFE) values. Mature miRNA sequences are highlighted in orange. The RNAStructure server (https://rna.urmc.rochester.edu/RNAstructureWeb/Servers/Predict1/Predict1.html; last accessed on 20 September 2023) [54], using a temperature of 299.15 degrees Kelvin (26 °C, a probable temperature for baculovirus-infected larvae), and default parameters, was employed for generating the structures. The base pair probabilities are indicated by the colors of the bases. This figure is original and has not been previously published.

**Figure 5 ijms-25-00603-f005:**
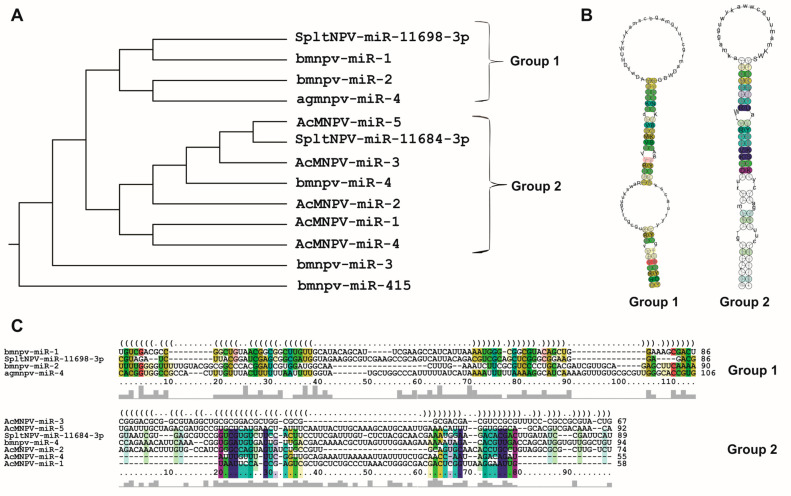
Conservation of secondary structures in baculoviral pre-miRNAs. The LocARNA-P server (http://rna.informatik.uni-freiburg.de/LocARNA/Input.jsp; last accessed on 20 September 2023) [55], with a temperature setting of 26 °C, was employed in this work to perform a comparative analysis of the secondary structures of miRNAs against experimentally determined structures. The generated guide tree encompasses all the sequences (**A**) and showcases hierarchical clustering to depict similar relationships, guiding the structural multiple alignments. This tree reveals the emergence of two distinct groups of structures that exhibit varying degrees of conservation (Group 1 and Group 2). With the identification of these two groups, the analysis was reiterated, utilizing only sequences belonging to each one (**B**,**C**), leading to the identification of structural conservation conforming to the typical pattern. Compatible base pairs are color coded to indicate sequence conservation by hue (C-G, G-C, A-U, U-A, G-U, or U-G) and structural conservation by saturation. This figure is original and has not been previously published.

**Table 3 ijms-25-00603-t003:** Baculoviral pre-miRNA endpoint stabilities. The ‘dot2ct’ and ‘efn2’ programs [Select Nucleic Acid Type: RNA; Temperature (K): 299.15; Write Thermodynamic Details File: select] on the RNAStructure server (https://rna.urmc.rochester.edu/RNAstructureWeb/; last accessed on 20 September 2023) [57] were utilized in this work to generate the secondary structures of baculoviral pre-miRNAs cleaved by Dicer and evaluate the thermodynamic details at both ends. This table is original and has not been previously published.

Bac-miR	Guide Strand (miRNA)		Passenger Strand	
	Exterior Loop	First Stack	Exterior Loop	First Stack
bmnpv-miR-1	−0.3	−1.1	−0.3	−2.5
bmnpv-miR-2	−2	−1.7	−0.2	−2.7
bmnpv-miR-3	−1.4	−2.8	−0.3	0
bmnpv-miR-4	−2	−3.7	−0.3	−1.1
BmNPV-miR-415	0.5	−2.5	−1.5	−2.5
AcMNPV-miR-1	−0.2	−2.7	−1.3	−3.7
AcMNPV-miR-2	−0.1	0	−0.1	−1.5
AcMNPV-miR-3	−1.9	−3.8	−0.3	−3.8
AcMNPV-miR-4	0.5	−1.1	−0.3	−2.4
AcMNPV-miR-5	−1.9	−2.8	−0.4	−1.2
agmnpv-miR-4	−1.9	−3.7	−0.1	0.5
SpltNPV-miR-11684-3p	0.4	−1.7	−0.7	−3.5
SpltNPV-miR-11698-3p	−0.4	−1.7	−0.5	−1.7

**Table 4 ijms-25-00603-t004:** Some bioinformatic tools for miRNA studies. * Type refers to the features of the tool: software (Soft) or database (DB). This table is original and has not been previously published.

Tool (Type) *	Details (Last Update)	URL	Reference
Mfold (Soft)	Secondary Structure Prediction (2003)	http://www.unafold.org/mfold/applications/rna-folding-form.php (accessed on 27 November 2023)	[59]
miRanda (Soft)	Target Prediction (2005)	http://cbio.mskcc.org/miRNA2003/miranda.html (accessed on 27 November 2023)	[60]
miRbase (DB)	General (2018)	https://www.mirbase.org/ (accessed on 27 November 2023)	[35]
miRDB (DB)	Target predictions and functional annotations (2019)	https://mirdb.org/ (accessed on 27 November 2023)	[61]
miRge3.0 (Soft)	Comprehensive analysis of small RNA sequencing data (2023)	https://github.com/mhalushka/miRge3.0 (accessed on 27 November 2023)	[62]
miRTarBase (DB)	Experimental microRNA-target Interactions (2021)	https://mirtarbase.cuhk.edu.cn/ (accessed on 27 November 2023)	[63]
miRTarget (Soft)	Target Prediction (2021)	https://rdrr.io/bioc/miRSM/man/miRTarget.html (accessed on 27 November 2023)	[64,65]
PITA (Soft)	Target Prediction (2008)	https://genie.weizmann.ac.il/pubs/mir07/mir07_prediction.html (accessed on 27 November 2023)	[66]
Rfam (DB)	Structural RNAs (2022)	https://rfam.org/ (accessed on 27 November 2023)	[67]
RNA22 (Soft)	Target Prediction (2016)	https://cm.jefferson.edu/rna22/Interactive/ (accessed on 27 November 2023)	[68]
RNAcentral (DB)	Non-coding RNAs (2023)	https://rnacentral.org/ (accessed on 27 November 2023)	[69]
RNAfold (Soft)	Secondary Structure Prediction (2021)	http://rna.tbi.univie.ac.at/cgi-bin/RNAWebSuite/RNAfold.cgi (accessed on27 November 2023)	[70]
RNAhybrid (Soft)	Target-microRNA hybridization prediction (2013)	https://bibiserv.cebitec.uni-bielefeld.de/rnahybrid (accessed on 27 November 2023)	[71]
TarBase (DB)	General (2018)	https://dianalab.e-ce.uth.gr/html/diana/web/index.php?r=tarbasev8%2Findex (accessed on 27 November 2023)	[72]
TargetScan (Soft)	Target Prediction (2018)	https://www.targetscan.org/vert_72/ (accessed on 27 November 2023)	[73]

## Data Availability

Not applicable.

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
