# Peer review of "A Comprehensive Study of MicroRNA in Baculoviruses"

_ijms, 2024, doi:10.3390/ijms25010603_

Round 1

Reviewer 1 Report

Comments and Suggestions for Authors

This manuscript is proposed as a review, but introduces new analysis that appears to me as original work. It proposed the development of a structured pipeline of sequence analysis. Accordingly, I am not sure that it should be published as "review".

The manuscript deals with baculovirus mi-RNAs, but I do not think it could be considered as claimed "a comprehensive review of miRNA biogenesis, function and characterisation in general" I do not think that the title is appropriate, as likely soon new work will be more comprehensive. 

My major concern is the need of clearly differentiating between data obtained only by sequence analysis and experimental evidence, that is differentiating between putative [promoters, terminators, mi-RNAs, structures] and experimental observations. Accordingly, it is importance to highlight the coherence (or the discordances) between the results from predictions and the experimental results.

Detailed comments

Line 69

Do you mean that independent sequence analysis gives exactly the same number of baculovirus species than ICTV? Quite logical as ICTV uses genetic distance to define a species.

However, what is the interest of this point? It would not be enough to say that there are 107 species approved by ICTV, and for some of them, the complete genome sequence of various isolates is available

Line 82

I would change the order: first the expanded knowledge, that could result in broaden applications

Line 108

What do you mean here? That looking only to "protein encoding genes" does not explain cellular behavior? and that exploring RNA genes might unveil regulatory mechanisms?

Line 123

“This conserved module encoded…” Not clear to me what "module" means here, neither what is the "conserved function" as they act in a variety of ways.

Line 130

"During this century" : could it not be better to say "since the beguining of the century? or from the 2000s ...

Line 144 (Figure 1).

In panel A it should be indicated that counts for 2023 are partial

Line 211

Change “these nucleic acids” by “their genomes” to refer to baculoviruses

Line 212

The authors indicate “at least 13 reported bac-miRs”. This is not clear. If the manuscript is a comprehensive review, it should be 13 reported, but not “at least”; If the authors wanted to state that more putative bac-miRs have been proposed, but only 13 have been experimentally verified, it should be clearly indicated, maybe indicating the total number of putative bac-mi-Rs deduced by sequence analysis.

Line 219.

It would be preferable to turn it like “this probably reflects the higher research effort devoted to this genus, partially due to the availability of host cell lines” instead of asking for more research efforts in other species

Line 227

Do you mean that potential miR sequences are present in all orthologs of all other baculovirus species?

Line 242

“this suggests that …” It is not clear to what “this” refers. Do the authors want to indicate that the absence of knowledge (experimental) make difficult to predict the actual size of many (putative) pri-miRNA ? or it is that the experimental data reveal that pri-miRNA are relatively long?

Line 244.

The paragraph is not clear enough about the data collected from literature, and the new contribution, and that of experimental results, and that deduced from sequence analysis.

Line 269. Table 1.

From the title of table 1, it is clear that the miRNAs in the list have been experimentally analysed, but it is not clear if the targets (viral or host) were also experimentally found. Please indicate “potential target” or “experimentally verified target”. I think that this could enhance the interest of the pipeline in the discussion.

Line 314

aiding to predict their existence?

Line 318

“Regarding this, the describes bac-miR were …” Is this your own original work?

Line 319

Attention, the legend of fig 3 is not in the right place.

Line 343

There are differences in the structures proposed here and those originally proposed by Sigh et al 2010. Could you discuss this point? What is the confidence on these interpretations?

Why they are organised like that: alphabetical order? discovery data?

Why not to group them according to the proposed group?

Line 357

It would be useful to discuss how characteristic are these two classes, by comparing to pre-miRNAs from other organisms, ie, the host.

Line 412

rather to say "predictions" or "estimates" instead of "measurements"

Line 429

I found this conclusion too affirmative. Having a good pipeline would allow finding putative bac-miRNAs. However, this would not result directly in a beter understanding of the molecular interplay

Line 462

“Analysis” instead of “observations?

 Line 663 

A relatively recent review (2020) by  Singh addresses the role of miRNA in the interactions of baculoviruses with their hosts 

Comments on the Quality of English Language

I found some paragraphs hard to understand. I indicated some in the review. I think sometimes it would be easier to write in present rather than in passive tenses.

Reviewer 2 Report

Comments and Suggestions for Authors

Motta et al provide here a nice introduction to the concept of miRNA in baculovirus genomes; specifically, in alphabaculovirus genomes (those of lepidopterans). The paper is well composed: the structure is logical and flows nicely, covering baculovirus basics clearly, then moving into the nature of (primarily small) regulatory RNAs like miRNA, before moving into baculovirus miRNAs specifically. The figures are useful and clear, adding to the communication quality of the paper, while the writing is clear, precise and accurately describes the science. The authors examine the 13 known alphabaculovirus miRNAs for conservation that could be predictive for identification and possibly understanding (eg function) novel miRNAs. While all of the paper is well done, ultimately it falls a little short in not extending the reviewed aspects to prediction. However, this is understandable given the review nature of the work and it settles as a useful resource for those interested in baculovirus (and other viruses’) potential miRNA manipulation of gene expression in host interactions. 
